# Research on Bathymetric Inversion Capability of Different Multispectral Remote Sensing Images in Seaports

**DOI:** 10.3390/s23031178

**Published:** 2023-01-19

**Authors:** Wei Shen, Jiaqi Wang, Muyin Chen, Lihua Hao, Zhongqiang Wu

**Affiliations:** 1School of Marine Science, Shanghai Ocean University, Shanghai 201306, China; 2Shanghai Estuary Marine Surveying and Mapping Engineering Technology Research Center, Shanghai 201306, China; 3School of Information Science and Technology, Hainan Normal University, Haikou 571158, China

**Keywords:** random forest mode, band ratio model, bathymetry inversion, multispectral imagery

## Abstract

In recent years, remote sensing has become an indispensable supplementary method for determining water depth in the seaports. At present, many scholars use multi-spectral satellite data to invert the water depth of the seaports, but how to select the appropriate satellite data in the seaports area is worth exploring. In this article, the differences in the retrieving ability between domestic and foreign multispectral images are compared, through building the random forest model and the band ratio model, which use different multispectral images to conduct retrieving water depth in Nanshan Port in conjunction with the WBMS multi-beam sounding system. The band ratio model and random forest model are chosen for water depth exploration, remote sensing images use GF-6, GF-2, Sentinel-2B, and Landsat 8 OLI data, which are all popular and easily accessible. The final experiment results from the constant adjustment of the model parameter show that the domestic series of GF-6 images performed the best in this experiment. The Root Mean Square Error (RMSE) and Mean Relative Error (MRE) of the random forest model are only 1.202 and 0.187, respectively. Simultaneously, it is discovered that the ‘Red Edge’ band of GF-6 is also very helpful in improving the accuracy of water depth inversion, which is rarely mentioned in previous studies. To some extent, the preceding studies demonstrate that it is possible to investigate water depth using common multispectral remote sensing images. In the case of some bathymetry inversion models or in some waters, the aforementioned study demonstrates that it is possible to examine the water depth using domestic remote sensing images that are superior to foreign multispectral images in terms of bathymetry inversion ability.

## 1. Introduction

The exploration of coastal areas has increased in frequency over the last ten years; the scope has expanded, and the means of exploration have become more diverse. However, all exploration activities, such as port construction, ocean bottom research, and the distribution of marine fish and algae, are based on water depth. Further exploration must be carried out with the goal of determining the water depth. Water depth can be measured in two ways: acoustic and optical. Acoustic methods can be used to obtain deeper and more accurate water depths, but they have a limited application scope and time. Remote sensing images have become an indispensable tool for determining coastal water depths.

Remote sensing has been used to determine water depth for about 50 years. As more satellites have been launched in recent decades, there has been an increase in the number of types of remote sensing images, which encourages the acquisition of remote sensing images from the side. Water depth development and detection through multi-spectral images have also advanced rapidly, and many scholars, both at home and abroad, have conducted extensive research. Lyzenga et al. [1] used multi-spectral aerial images for the first time to investigate the water depth of shallow sea areas. They developed a linear model for the inversion of water depth in a single band, introducing a new concept of water depth inversion; Stumpf et al. [2] proposed the bathymetric inversion model has too many parameters, so a band ratio model is proposed to solve the problem. There are only two unknown parameters in this model. Experiments on the model yielded very good inversion accuracy; Wang Yanjiao [3] considered the comprehensive factor of bathymetry inversion, explored the linear and nonlinear relationship between bathymetry and remote sensing image spectrum, and partially solved the problem of bathymetric information acquisition in near-shore turbid waters. Spitzer et al. [4] developed a water depth inversion algorithm for various remote sensing platforms using the bi-directional flow radiation transmission model, with spectral signals, seawater scattering characteristics, and low-quality types as input parameters, as well as the band ratio and multi-band model. Model-based inversion of water depth has limitations, and each model is only applicable to a specific water area. To be applicable in other bodies of water, the parameters of this model must be modified [5]. William Hernandez investigated the effect of atmospheric correction on the accuracy of water depth detection in WorldView-2 images using dark subtraction and the Cloud Shadow Approach (CSA). When a good atmospheric correction method is used and the seabed sediment changes or approaches the limit of remote sensing depth, the depth can be obtained using a WorldView-2 image. Xiaohan Zhang [6] has been validated in three regions by replacing measured water depth points with ICESat-2 water depth points. They all produced satisfactory results. The limitation of the measured water depth points can be reduced to some extent using ICESat-2 water depth points.

As computer technology advances, machine learning and deep learning models are being applied to water depth inversion. The random forest model is widely used in various fields as an excellent machine learning model, and many scholars have also conducted extensive research on water depth acquisition in shallow waters. Qiu Yaowei et al. [7] compared the water depth obtained by random forest inversion to the water depth obtained by semi-empirical and semi-theoretical models. They discovered that the water depth obtained by the random forest model was superior to the water depth obtained by semi-empirical and semi-theoretical models. Mateo-Pérez et al. [8] used a random forest model and RBF-kernel SVM technology to obtain more accurate depth prediction in Candás Port, and discovered that the obtained accuracy has a strong relationship with the selected control points. The water area of the port is relatively turbid, which has a good promotion effect on the water depth inversion of the port water area by remote sensing. Lukasz Janowski et al. [9] used airborne lidar images to identify 9 types of seabed landforms and 3 types of man-made structural landforms, with a classification accuracy of 94%. This method is helpful to improve the accuracy of water depth inversion near the seaports. We can further improve the accuracy by determining the influence of seabed sediment on water depth inversion. Diosantos et al. [10] measured the underwater landform of 220 km on the west coast of Portugal with the fast Fourier transform and wavelet method using Sentinel-1 data and synthetic aperture radar data. At the same time, they discovered that the slope of the seabed has a significant impact on accuracy, with the steeper the seabed, the greater the error. The sea area near the seaport is generally flat, which is favorable to water depth inversion. Liu Kai et al. [11] used the skeleton interpolation method and the XGBoost model to predict the depth of unknown underwater areas. After testing twelve representative lakes on the Qinghai-Tibet Plateau, both methods can obtain a more accurate water depth, but the XGBoost model performs better overall.

Bathymetry research using remote sensing is extensive. At first, there are numerous image data sources to choose from. Bathymetry makes use of a variety of multispectral and hyperspectral data, as well as lidar and synthetic aperture radar images. Second, remote sensing sounding models are becoming increasingly diverse, progressing from the initial linear model, to the semi-empirical and semi-theoretical models, to the current machine learning model. Many scholars concentrate on remote sensing sounding based on vertical comparison, and inversion accuracy is improved further, but inversion accuracy is poor with horizontal comparison. In this paper, some sea areas of Sanya’s Nanshan Port are chosen for experiments. Water depth inversion is performed using domestic images of GF No.2 and GF No.6, Landsat8 OLI images, and Sentinel-2B images from abroad, and the differences in water depth inversion capabilities of remote sensing images from different platforms are discussed, providing a certain reference significance for selecting remote sensing images for later water depth acquisition.

## 2. Study Area and Data

### 2.1. Study Area

This experiment was conducted in a portion of the sea area (18°18′30″ N, 109°7′00″ E) of Nanshan Port, Yazhou District, Sanya City, Hainan Province. As shown in Figure 1. Although the water depth does not exceed 20 m, the water quality in this area is poor. Therefore, it has a certain reference value for the extensive practice of obtaining water depth using remote sensing images. 

### 2.2. Measured Water Depth Data

Two acoustic measurements were taken in the sea area of Nanshan Port from 11 July to 13 July 2021. The Norwegian NORBIT Company’s WBMS multi-beam sounding system was used to measure the water depth in the experimental area by ship. The WBMS multi-beam bathymetry system operates on the principle of using a transmitting transducer array to transmit an acoustic wave, covering a wide sector to the seabed and a receiving transducer array to receive an acoustic wave with a narrow beam, as shown in Figure 2. The ‘footprints’ of the seabed topography are formed by the orthogonality of the transmitting and receiving sectors, and these ‘footprints’ are properly processed. One detection can provide the water depth value of hundreds or more seabed measured points in the vertical plane perpendicular to the heading, allowing the underwater target’s size, shape, and height change to be accurately and quickly measured within a certain width along the route. More accurately, it depicts the three-dimensional characteristics of seabed topography. When combined with on-site navigation, positioning, and attitude data, high precision and high-resolution digital results can be obtained [12].

The wind speed is between 2–3 levels, and the wave height is between 0.1–0.3 m during the measurement process. These conditions satisfy the fundamental requirements of satellite sounding. Tidal offset correction in the study area is based on data obtained from the China Maritime Safety Administration’s website, in order to make the image data as consistent as possible with the measured data and to ensure the visibility of the image. The final correction of sounding results is based on tidal data, as shown in Figure 3. The experimental data were processed to yield 3 million measured water depth points and 4000 data points extracted from the experimental area. The extracted data points were evenly distributed in the experimental area to improve the accuracy of water depth inversion as much as possible, with 3200 points used as training sets and 800 points used as test sets.

### 2.3. Image Data

The GF-6 WFV wide-format image and GF-2 PMS image data used in the experiment were downloaded from the China Resources Satellite Center’s “Land Observation Satellite Data Service” (http://data.cresda.com:90/#/mapSerach, access on 5 December 2022). The cloud cover is as small as possible. The date of the GF-6 image is 19 June 2021 and of GF-2 image is 16 January 2021. The Landsat 8 OLI data was downloaded from the geospatial data cloud platform (http://www.gscloud.cn/search, access on 5 December 2022), and the imaging date of the Landsat 8 OLI image is 03 December 2021; Sentinel-2B data was downloaded from the official website of the US Geological Survey (https://www.usgs.gov/, access on 5 December 2022), and the imaging time was 16 June 2021. All of the images used in the experiment are multispectral, all of the image information is shown in Table 1. During the experiment, the visible light band is primarily used to determine the depth of the water. The goal of selecting remote sensing images is to stay as close to the measured data time as possible while maintaining image quality. Because the image visibility in the study area should be as high as possible and the cloud cover is less than 3%, the final imaging time of some images differs significantly from the time of the measured data. The four image data used in the experiment are common multispectral remote sensing images, which are widely used in various fields such as agricultural monitoring and atmospheric environmental monitoring, and have relatively mature applications in many fields.

## 3. Research Methods

### 3.1. Experimental Process

After acquiring the remote sensing image data, we performed radiometric calibration, which is the process of converting the digital value (DN, Digital Number, also known as digital quantization value) recorded by the sensor into absolute radiance using a rational function model [13]. The process of converting the radiance at the top of the atmosphere to surface radiance is known as atmospheric correction. The primary goal of atmospheric correction is to reduce atmospheric scattering and reflected energy received by satellite sensors, while obtaining as much real surface reflectivity information as possible. The FLAASH model is used in the experiment for atmospheric correction. There are numerous advantages to using the FLAASH atmospheric correction module. (1) It is very friendly to multi-spectral remote sensing data and can process multi-spectral data well; (2) The algorithm is highly accurate; the model employs the MODTRAN5 radiative transfer model, which is more accurate than the look-up table method. (3) Different correction models are available for different regions, and the corresponding atmospheric model and aerosol model can be chosen based on the region, etc. [14]. Therefore, the FLAASH module is used in this paper to perform atmospheric correction on satellite images. The spectral curve of the same water area before and after atmospheric correction is shown in Figure 4.

The rational function model (RFM) is used for geometric correction to correct the geometric error caused by the coordinate shift on the remote sensing image, and make the pixel points better match the measured data points. The strict imaging model is a new sensor imaging model with nearly the same accuracy, allowing geometric correction to be performed using rational function model coefficients provided by satellite manufacturers. Water and land separation can reduce the influence of land area on water inversion and improve water information, making it easier for inversion models to obtain water information from images. The process of this experiment is formulated according to the above content, using the random forest model as an example, as shown in Figure 5 below.

### 3.2. Inversion Mode

The band ratio model and the random forest model have always performed well in water depth inversion when combined with the research of many scholars [15,16,17,18]. Because the requirements are less stringent than for other inversion models, these two models were chosen for bathymetric acquisition.

(1)Band ratio model

The band ratio model is based on the single-band model and takes into account seabed substrate changes, assuming that the reflectivity ratio of the two bands in different seabed substrates is constant. The influence of water body properties and water bottom type changes on the accuracy of water depth inversion can be reduced to some extent using this calculation method. At the same time, it can eliminate or suppress the influence on water depth inversion caused by changes in satellite attitude, the angle of the sun’s altitude, the wave on the water’s surface, and the scanning angle. The blue-green band with relatively strong water penetration is usually chosen from the two bands [19]. 

(2)Random Forest Model

A machine learning algorithm that integrates decision trees is known as the random forest. It combines the benefits of methods such as bagging and random selection of feature splitting, as well as the addition of two random quantities—random selection of samples and random selection of features [20]. Therefore, the random forest algorithm has numerous advantages. The random forest model’s generalization error tends to an upper bound as the number of decision trees increases, so it has the advantage of preventing overfitting; in the model’s calculation process, the selection of training samples and split attribute sets are performed randomly, so the model can be built independently without too much manual intervention, and the process is very simple. The optimal splitting formula is shown in formula (1); each decision tree between the models can be processed in parallel. The outliers appear to have a high tolerance as well. Finally, we can get the prediction results of this input data set on the entire random forest by using Formula (2) to average the predicted values on each decision tree. However, because the random forest model is highly dependent on the training set, the training set must be used carefully during the experiment, and different parameters should be tried to obtain the best inversion results [21].
(1)$$\frac{I}{N_{l}}\sum a{\left(\left|C_{a,l}\right|\right)}^{2}+\frac{I}{N_{r}}\sum a{\left(\left|C_{a,r}\right|\right)}^{2}$$
(2)$$p(c|v)=\sum_{t=i}^{T}P_{t}(c|v)$$

In the formula, $N_{l}$ represents the total number of left split samples, $N_{r}$ represents the total number of right split samples, $C_{a,l}$ represents the number of predicted value a in right split samples; $C_{a,r}$ represents the number of samples of the predicted value an in the left splitting, T represents the total number of trees generated in the prediction process, c is a specific value, and P represents a probability function.

### 3.3. Accuracy Evaluation

The root mean square error is used in the marine mapping. RMSE is commonly used to assess the precision of the observed or inverted value. It is the square root of the ratio of the square sum of the deviation between the observed value and the true value, and the ratio of the number of observations n, as shown in Formula (3). The lower the RMSE in the result analysis, the higher the accuracy of the inversion results.

The average value of the relative error is referred to as MRE. The formula for calculation is Formula (4). The absolute value, that is, the absolute value of the average error, is commonly used to express this average relative error. The smaller the MRE, the smaller the range of deviation from the true value in the result analysis. 

Comparing the fitting effect of bathymetry data inverted by different models and corresponding measured bathymetry data using the correlation coefficient R^2^ [22].
(3)$$RMSE=\sqrt{\frac{1}{n}\sum_{i-1}^{n}{\left(y_{i}-P_{i}\right)}^{2}}$$
(4)$$MRE=\frac{1}{n}\sum_{i-1}^{n}|\frac{y_{i}-P_{i}}{yi}|*100\%$$
(5)$$R^{2}=\frac{\sum_{i=1}^{n}{\left(y_{i}-y\right)}^{2}-\sum_{i=1}^{n}{\left(y_{i}-P_{i}\right)}^{2}}{\sum_{i=1}^{n}{\left(y_{i}-y\right)}^{2}}$$

In the formula, $y_{i}$ represents the measured water depth of the *i*-th test point, $P_{i}$ represents the inversion water depth of the *i*-th sample point, y represents the average measured water depth of the test point, and n represents the number of test points.

This paper uses the root mean square error RMSE and the average relative error MRE as quantitative evaluation indicators to evaluate the accuracy of model inversion, in order to objectively evaluate the inversion accuracy of each model. The lower the MRE and RMSE values, the more accurate the bathymetric inversion and the better the inversion model’s fitting effect. The model’s inversion accuracy is determined by calculating the difference between 800 data points in the measured water depth and the inversion water depth value at the same location.

## 4. Experimental Result

The water depth inversion is performed using the processed remote sensing images, the random forest model, the band ratio model, and the sounding points obtained by the WBMS multi-beam bathymetry system, and the root mean square error (RMSE) and mean relative error (MRE) of different remote sensing images are calculated. The final results are shown in Table 2 below.

Table 2 shows that the overall quality of the random forest model’s inversion results is higher than that of the band ratio model’s inversion results. The final water depth map is shown in Figure 6 below, after reclassifying the random forest model’s inversion results and removing or merging the abnormal pixel values. In Figure 7, the color changes from red to green, indicating that the point’s depth changes from shallow to deep. The results obtained by the random forest model and the band ratio model are also quite different, as shown in Figure 7. The water depth measured by the band ratio will exhibit a ‘fault’ phenomenon. At 4.5–5 m, the GF-6 image has no water depth result, whereas the Sentinel-2B data has no water depth result at about 4 m. The other two images depict similar scenarios.

## 5. Discussion

When comparing the inversion result map to the water depth map of our measured data, Figure 3, it is easy to see that there is a channel about 10 m deep in the actual experimental area, but none of the inversion result maps represent this channel. Therefore, there will be some details missing when obtaining water depth from the remote sensing images. We intercepted the contour map of the study area on the public electronic chart website (https://webapp.navionics.com, access on 5 December 2022) for a more obvious comparison. In Figure 8, the shallow color indicates that the water depth value is large. It is clear that only the GF-6 inversion results follow the same trend as the electronic chart depth change, while the other inversion results vary slightly.

We use various methods to measure the root mean square error of various water depth ranges using different images, and we can see that the overall trend of the root mean square error is of the ‘U’ type, as shown in Figure 9. This is because when the water depth is too shallow, within 0–3 m, the seabed sediment reflection and water mixture seriously affect the accuracy of water depth inversion; when the water depth is too deep, above 6 m, the absorption of light by water also increases, causing serious loss of spectral information and reducing the accuracy of water depth inversion. Between 3–6 m, the water absorption of light and the seabed, water mixture to achieve a ‘neutralization value’, and the water depth inversion effect are optimal. This phenomenon is also reflected in Figure 7: when the depth is less than 3 m or greater than 6 m, the prediction point deviates from the regression line more clearly, and when the depth is 3–6 m, the prediction points are close to the regression line. The outcomes of studying various images include some details that must be considered. When the water depth is 5 m, the high-resolution series images and LandSat 8 OLI data all achieve the minimum root mean square error, while the Sentinel-2B image achieves the minimum when the water depth is 4 m. As can be seen, the depth of different images to achieve the best performance varies.

When the evaluation indicators are compared, it is clear that the RMSE obtained by the GF-6 image in the results obtained using the random forest model is only 1.202, and the MRE is only 0.187. The image with the highest ground resolution in this experiment is GF-2, which is also a high-scoring series, and its inversion results achieve high accuracy, but its final performance is not as good as the other three remote sensing images. In this experiment, the accuracy of the results obtained by our domestic GF-6 image outperformed the foreign Sentinel-2B and Landsat 8 OLI data, and it is the best result in this water depth inversion experiment.

The GF-6 satellite is China’s first with a “Red Edge” band sensor, providing critical remote sensing data support for agricultural monitoring and development. The “Red Edge I” band is B5, and the “Red Edge II” band is B6. Separate tests are performed on the two “Red Edge” bands. The “Red Edge” band is excluded from the calculation of water depth detection using the random forest model. The results in the following Table 3 also show that when there is no “red edge” band to participate in the calculation, the inversion accuracy is still significantly reduced, with the RMSE being 1.497 m, only 0.027 m higher than the worst GF-2 image, and the MRE being 0.28, the worst among the multispectral images used. When only one “Red Edge” band is used in the calculation, the inversion accuracy improves noticeably. Therefore, the two “Red Edge” bands of GF-6 can not only improve crop classification accuracy, but also greatly improve water depth inversion accuracy.

Sentinel-2B has a relatively high ground resolution among the multispectral data used in the experiment, but its inversion result is not as good as the GF-6 image. It is speculated that this is because we only use three Sentinel-2B bands, red, green, and blue, and do not fully utilize all bands during the inversion process. Therefore, only the red, green, and blue bands of the GF-6 image are used for water depth inversion, and the Sentinel-2B inversion results are compared, as shown in Table 4 below. When only three bands are used for water depth inversion, the GF-6 inversion results do not match those of Sentinel-2B. This demonstrates that the GF-6 ‘red edge’ band is extremely important in improving the accuracy of water depth inversion.

## 6. Conclusions

Four different types of multi-spectral images are used in this paper to calculate water depth using two commonly used water depth inversion models: the random forest model and the band ratio model. In a shallow area of 20 m near the coast, these multi-spectral image soundings have significant capabilities. The average relative error (MRE) is less than 20%, and the root mean square error (RMSE) is less than one meter. However, depending on the depth, the accuracy of water depth inversion varies greatly. The water depth detection results are very good in the range of 3–6 m, but the inversion accuracy is significantly reduced in the range of less than 3 m and more than 6 m. Simultaneously, it was discovered for the first time that the GF-6 image’s ‘Red Edge’ band is very useful in improving the accuracy of water depth inversion. Additionally, we further discovered that the “Red Edge” band of the GF-6 image is very useful in improving the accuracy of water depth inversion, but the specific effects and how they are affected need to be studied further. In this experiment, the random forest model significantly outperforms the band ratio model. The water depth inversion model outperforms the traditional regression model, represented by the random forest machine learning model in terms of effect and generalization ability. It can be used to detect water depth near the coast.

To summarize, existing multispectral images and widely used water depth inversion models are extremely effective at detecting water depth. In this paper, four common multispectral remote sensing images for water depth inversion are compared, and it was discovered that the GF-6 image performs best in this experiment. However, there have been few previous studies that compare the water depth inversion of various remote sensing images. Therefore, this paper provides a relatively simple technical process for scholars for reference. The water depth detection capability of GF-6 images has improved with the advancement of domestic remote sensing images, to the point where it can even outperform foreign excellent multispectral images. Although remote sensing sounding has advanced, inversion accuracy still needs to be improved. Future research will focus on optimizing and innovating inversion models, as well as improving image preprocessing methods.

## Figures and Tables

**Figure 1 sensors-23-01178-f001:**
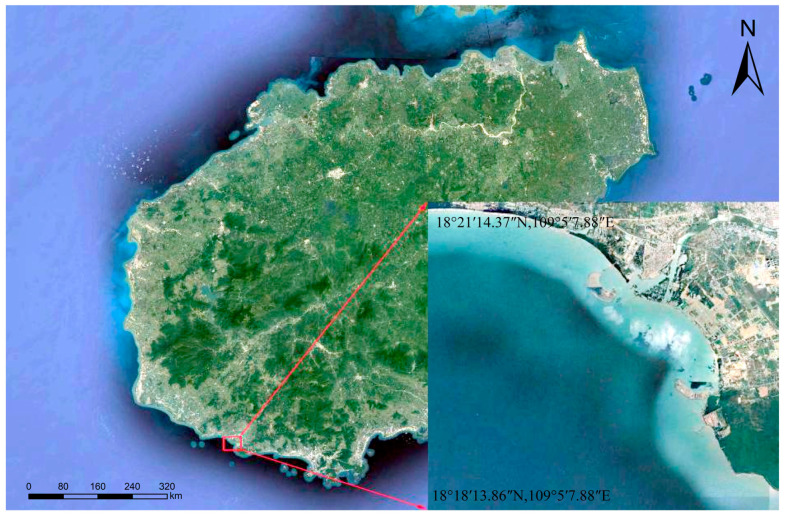
Location Map of Nanshan Port.

**Figure 2 sensors-23-01178-f002:**
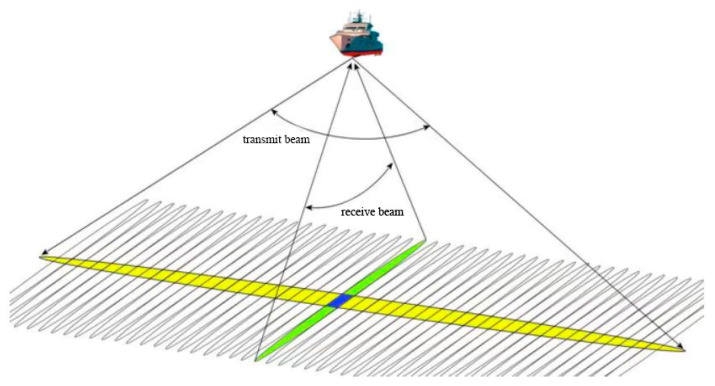
WBMS sounding principle diagram.

**Figure 3 sensors-23-01178-f003:**
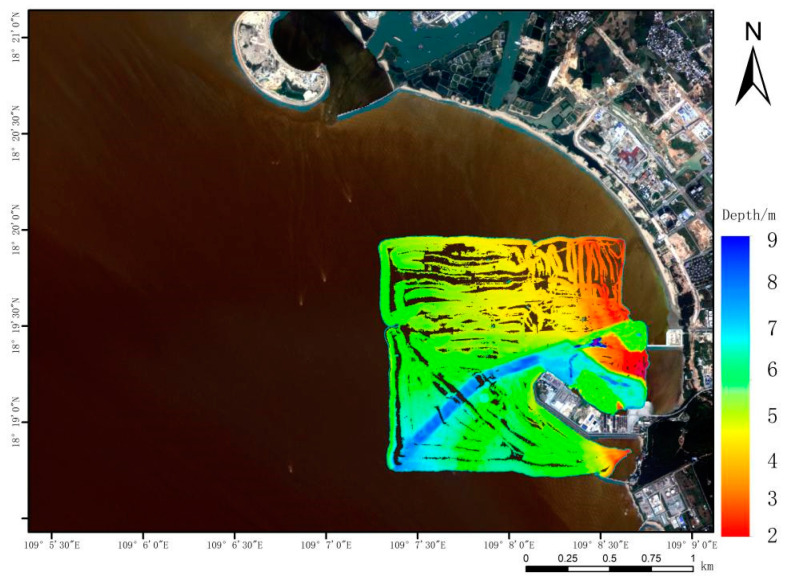
Measured water depth map of Nanshan Port.

**Figure 4 sensors-23-01178-f004:**
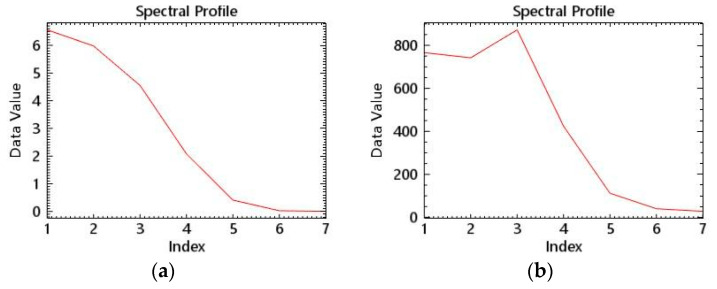
Spectral curves before and after atmospheric correction. (**a**) Before atmospheric correction. (**b**) After atmospheric correction.

**Figure 5 sensors-23-01178-f005:**
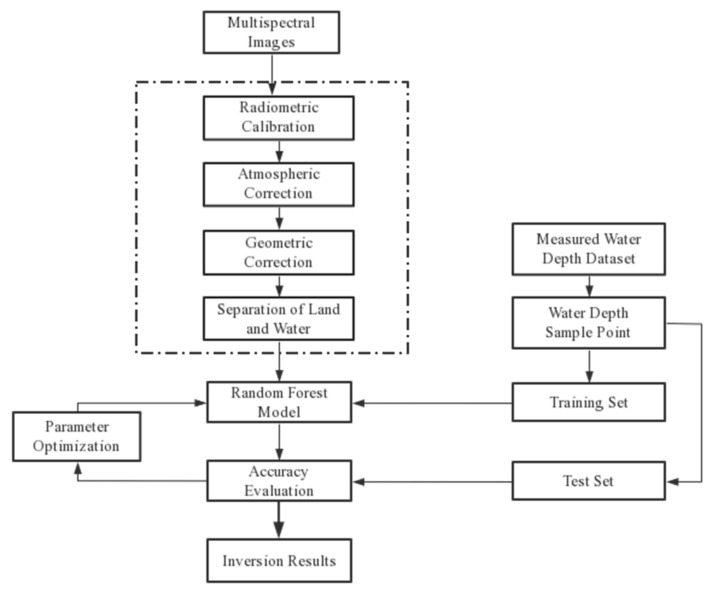
Flow chart of multi-spectral remote sensing data water depth inversion technology.

**Figure 6 sensors-23-01178-f006:**
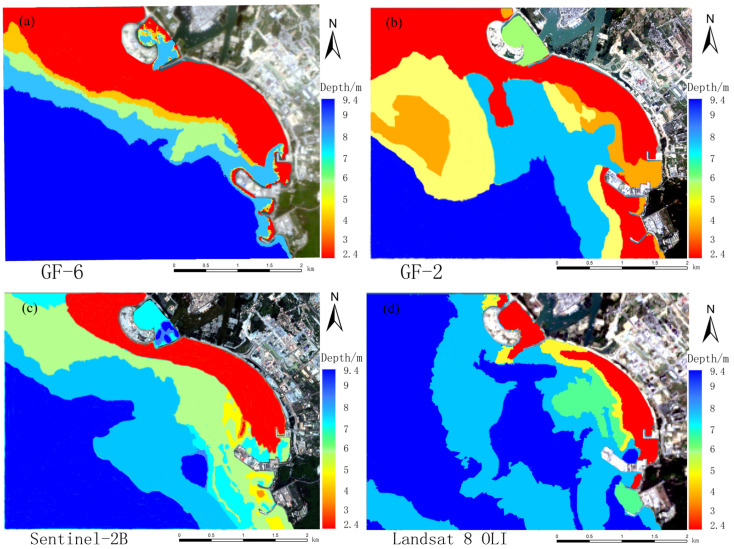
Inversion of water depth map by random forest model of a multispectral image: (**a**) GF-6; (**b**) GF-2; (**c**) Sentinel-2B; (**d**) Landsat 8 OLI.

**Figure 7 sensors-23-01178-f007:**
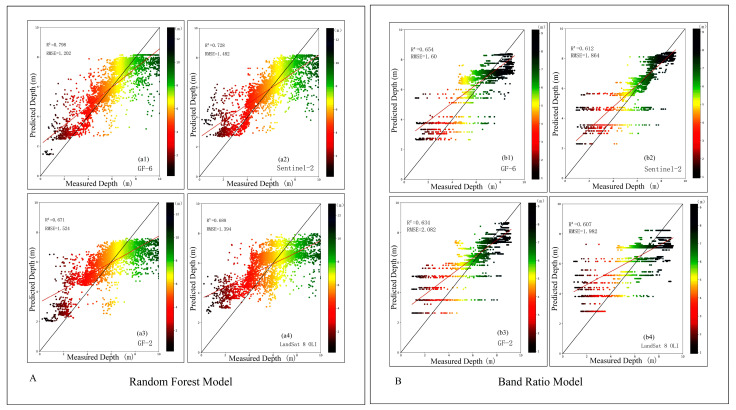
Correlation between water depth inverted by different images of different models and measured water depth: (**A**) is Random Forest Model, (**a1**) GF-6, (**a2**) Sentinel-2B, (**a3**) GF-2, (**a4**) Landsat 8 OLI; (**B**) is Band Ratio Model, (**b1**) GF-6, (**b2**) Sentinel-2B, (**b3**) GF-2, (**b4**) Landsat 8 OLI.

**Figure 8 sensors-23-01178-f008:**
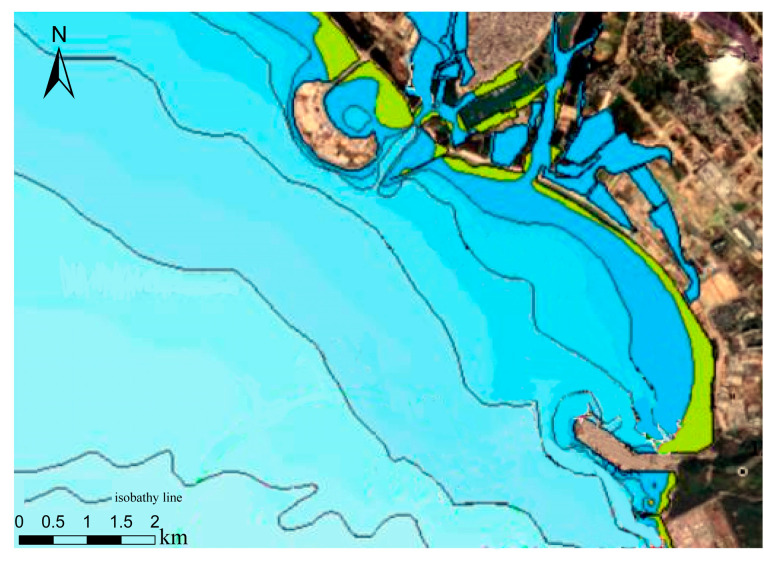
Isobath map of Nanshan Port waters.

**Figure 9 sensors-23-01178-f009:**
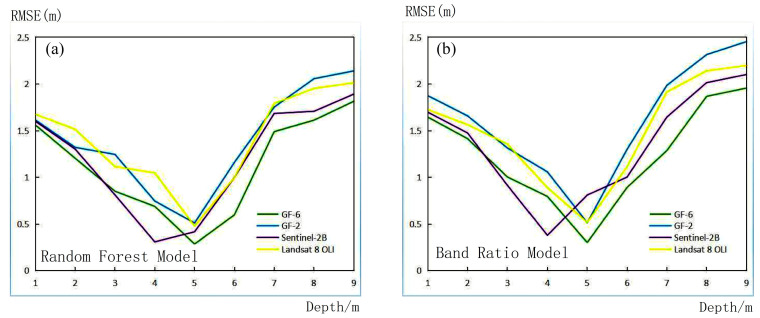
RMSE by different methods, different images, and different water depth range: (**a**) Random Forest Model; (**b**) Band Ratio Model.

**Table 1 sensors-23-01178-t001:** Information about image data.

Image	Spectral Range	Spatial Resolution	Width	Revisit Cycle
GF-6 WFV	0.45–0.90 µm/8 bands	≤16 m	≥800 km	5/69 days
GF-2 PMS	0.45–0.90 µm/5 bands	3.2 m	45 km	4/41 days
Sentinel-2B	0.442–21.85 µm/13 bands	**10**/20/60 m	100 km	10 days
Landsat 8 OLI	0.43–12.51 µm/11 bands	15/**30**/100 m	185 km	16 days

**Table 2 sensors-23-01178-t002:** Bathymetric inversion results from different multispectral images.

Inversion Model	Random Forest Model			Band Ratio Model		
Image	Band Selection	RMSE/m	MRE	Band Selection	RMSE/m	MRE
GF-6	All	**1.202**	**0.187**	B1 B2	**1.605**	**0.289**
GF-2	All	* 1.524 *	* 0.275 *	B2 B3	* 2.082 *	0.419
Sentinel-2B	B2 B3 B4	1.482	0.274	B2 B3	1.864	0.373
Landsat 8 OLI	All	1.394	0.240	B2 B3	1.982	* 0.431 *

**Table 3 sensors-23-01178-t003:** Inversion Results of Water Depth Affected by “Red Edge” Waveband.

Band Selection	RMSE/m	MRE
No B5	1.362	0.249
No B6	1.354	0.256
No B5/B6	1.497	0.280
All Involved	1.202	0.187

**Table 4 sensors-23-01178-t004:** ‘red green blue’ band water depth inversion results.

Inversion Model	Random Forest Model		Band Ratio Model	
Image	RMSE	MRE	RMSE	MRE
GF-6	1.493	0.278	1.905	0.395
Sentinel-2B	1.482	0.274	1.864	0.373

## Data Availability

The GF-6 WFV wide-format image and GF-2 PMS image data are from the (http://data.cresda.com:90/#/mapSerach, accessed on 5 December 2022); The Landsat 8 OLI data was downloaded from the (http://www.gscloud.cn/search, accessed on 5 December 2022); Sentinel-2B data was downloaded from the (https://www.usgs.gov/, accessed on 5 December 2022); contour map from the (https://webapp.navionics.com, accessed on 5 December 2022).

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
