# Peer review of "Research on Bathymetric Inversion Capability of Different Multispectral Remote Sensing Images in Seaports"

_sensors, 2023, doi:10.3390/s23031178_

Round 1

Reviewer 1 Report

Should be add the route of research technology.

Explanation the applicability and superiority of remote sensing imagery.

Whether to consider other imaging factors for the port pool depth of the remote sensing image process.

Expansion the  conclusions and  discussion.

Condense the innovation of the thesis

Author Response

Please see the file attached.

Reviewer 2 Report

In this paper, several different types of multi-spectral images are used for water depth inversion. The experimental results show that the inversion effect is good in the shallow water area, but it is not obvious in the place where the water ratio is too deep. In general, it is still effective to use multi spectral images to detect water depth, which can provide some reference for some practical applications.

However, this paper mainly describes the basic implementation of the method, and there is not much discussion about the theoretical basis behind the method. In order to provide more powerful theoretical support for the efficient application of practical projects, I wonder if you can strengthen it? 

Author Response

Please see the file attached.

Reviewer 3 Report

Review report for paper

technical note entitled “Research on Bathymetric Inversion Capability of Different Multispectral Remote Sensing Images in Seaports ‘’

General remarks:

·      Extensive editing of English language required

·      The references cited in the introduction section are relevant but should be updated

·      Quality of all figures should be improved (600dpi is proposed)

Specific remarks

·      The chart of methodology section should be explained and some important details are missing such as validation and accuracy evaluation.

·      The choice of the dates of image acquisition June, January and December should be explained in the text.

·      The image correction part (radiometric, atmospheric and geometric) should be more explained and illustrated by figures.

·      In the presentation of the study area, climatic data and description should be added because this can have a high influence on the water depth.

·      Figure 1 and Figure 7: maps are missing the North arrow, the scale, the legend and coordinates

·      Figures 5 and 6 are not clear

Author Response

Please see the file attached.

Reviewer 4 Report

Please see the file attached.

Author Response

Please see the file attached.

Round 2

Reviewer 3 Report

I would like to thank the authors for the correction except for a few remarks: - the text in Figures 6 and 7 is unreadable, increase the size and the resolution of figures

Author Response

Dear Professor:

Thank you very much for taking the time to review my manuscript again.

The picture has been modified or enlarged, thanks for the correction

Reviewer 4 Report

The authors have addressed all my comments. 

Author Response

Dear Professor:

Thank you very much for taking the time to review my manuscript again.

Thank you for your help. I'm very grateful.